# Cytokine-Mediated Dysregulation of Signaling Pathways in the Pathogenesis of Multiple Myeloma

**DOI:** 10.3390/ijms21145002

**Published:** 2020-07-15

**Authors:** Sabah Akhtar, Tayyiba A. Ali, Ammara Faiyaz, Omar S. Khan, Syed Shadab Raza, Michal Kulinski, Halima El Omri, Ajaz A. Bhat, Shahab Uddin

**Affiliations:** 1Translational Research Institute, Academic Health System, Hamad Medical Corporation, Doha 3050, Qatar; sabahaktr@gmail.com (S.A.); tayyiba1991@gmail.com (T.A.A.); Afaiyaz4825@gmail.com (A.F.); Mkulinski@hamad.qa (M.K.); 2Chicago College of Osteopathic Medicine, Midwestern University, Downers Grove, IL 60515, USA; Omarshahabkhan@gmail.com; 3Department of Stem Cell Biology and Regenerative Medicine, Era University, Lucknow 226003, India; drshadab@erauniversity.in; 4National Cancer Care and Research, Hamad Medical Corporation, Doha 3050, Qatar; Helomri@hamad.qa; 5Translational Medicine, Research Branch, Sidra Medicine, Doha 26999, Qatar; abhat@sidra.org; 6Dermatology Institute, Department of Dermatology and Venereology, Hamad Medical Corporation, Doha 3050, Qatar

**Keywords:** multiple myeloma, hematological malignancies, signal transduction, proliferation, cytokines

## Abstract

Multiple myeloma (MM) is a hematologic disorder of B lymphocytes characterized by the accumulation of malignant plasma cells (PCs) in the bone marrow. The altered plasma cells overproduce abnormal monoclonal immunoglobulins and also stimulate osteoclasts. The host’s immune system and microenvironment are of paramount importance in the growth of PCs and, thus, in the pathogenesis of the disease. The interaction of MM cells with the bone marrow (BM) microenvironment through soluble factors and cell adhesion molecules causes pathogenesis of the disease through activation of multiple signaling pathways, including NF-κβ, PI3K/AKT and JAK/STAT. These activated pathways play a critical role in the inhibition of apoptosis, sustained proliferation, survival and migration of MM cells. Besides, these pathways also participate in developing resistance against the chemotherapeutic drugs in MM. The imbalance between inflammatory and anti-inflammatory cytokines in MM leads to an increased level of pro-inflammatory cytokines, which in turn play a significant role in dysregulation of signaling pathways and proliferation of MM cells; however, the association appears to be inadequate and needs more research. In this review, we are highlighting the recent findings on the roles of various cytokines and growth factors in the pathogenesis of MM and the potential therapeutic utility of aberrantly activated signaling pathways to manage the MM disease.

## 1. Introduction

Multiple myeloma (MM) is an ailment of the plasma cells (PCs) characterized by the uncontrolled proliferation of long-lived monoclonal PCs. These PCs accumulate in the bone marrow, which causes impairment of bone strength and weakness of the immune system [1]. MM is the second most prevailing hematological malignancy after non-Hodgkin lymphoma, responsible for approximately 20% of deaths caused by hematological malignancies [2]. The disease is less common in women than men, and despite substantial improvement over the past decade in cancer therapeutics, myeloma cases and death rates have increased from 1990 to 2016 [3]. The average age of diagnosis is 66 years, and the five-year survival rate is 46.6%. The incidence of disease also differs in different ethnicities and is more common in Caucasians than in Asians. Although some patients survive a decade after diagnosis, most of them die within 24 months due to the progression of treatment resistance. Even though many novel chemotherapeutic drugs have been discovered and used to cure MM, the disease remains incurable due to the reduced response rate and toxicity of these drugs [4]. 

Active MM is supported by the bone marrow (BM) microenvironment. The growth and survival of MM clones are highly dependent on systemic cytokines [5]. Cytokines are a type of growth factors that regulate the balance between cell-based and humoral immune responses [6]. The bone marrow stromal cells (BMSCs) that are present in the MM niche produce considerable quantities of TGFβ and IL-6,7 and 8, which maintain the pro-tumorigenic conditions, regulate growth and survival of cancerous cells and maintain feedback loops of cytokines [7,8]. The autocrine production of cytokine IL-15 is shown to be involved in the survival of MM cells [9]. 

MM cells and BMSCs induce autocrine or paracrine secretion of numerous mediators [10]. BM microenvironment in MM contains high levels of IL-6, HGF, EGF, IL-2R and cytokines stimulated due to interferon-γ (IFN-γ) [11]. A number of these cytokines play a vital role in MM development by acting as growth factors of MM cells and promote cellular adhesion. There are some cytokines which are involved in osteoclastogenesis and angiogenesis [12,13,14,15]. The production of cytokines by subsets of T-lymphocytes and plasma cells in BM promotes the growth of malignant cells [10]. 

The growth of neoplasia is associated with inflammation, and an increase in pro-inflammatory cytokines can promote the growth of the tumor [16]. Cytokines are involved in both pro-inflammatory and anti-inflammatory processes [10]. The balance between chemokines and cytokines is a critical process in tumor induction. The inflammatory infiltrate, which is formed in a tumor, is highly dependent on cytokine balance. Tumors that produce few or no cytokines or those tumors that produce anti-inflammatory cytokines have limited growth of the tumor due to constrained inflammation and vascular responses. On the other hand, increased production of pro-inflammatory cytokines causes angiogenesis, thus support tumor growth [17]. 

## 2. Bone Marrow Microenvironment in MM

The BM milieu is composed of hematopoietic and nonhematopoietic cells; the extracellular matrix (ECM) and soluble components such as cytokines, growth factors and adhesion molecules [18]. BM microenvironment plays a critical role in the development of a disease. It is composed of various proteins of the ECM, including laminin, collagen, fibronectin, osteopontin and some cellular components, such as erythrocytes, hematopoietic stem cells, endothelial cells of bone marrow, osteoclasts, osteoblasts and immune cells (Figure 1). MM cells are attracted to BM through secretion of different cytokines (IL-6, BAF, IGF-1, FGF and SDF-1) and chemokine (CXCL-12) from these cellular components (Figure 1) [19]. There are various adhesion molecules, including ICAM, NCAM, CD40, VLA 4, VLA 5 and LFA 1, expressed in both BMSCs and myeloma cells. The interactions of these adhesion molecules result in the upregulation of several intracellular signaling pathways; for example, phosphoinositide 3-kinase (PI3K), signal transducer and activator of transcription 3 (STAT3), nuclear factor-kappa-B (NF-κB) and mitogen-activated protein kinase (MAPK), which leads to the secretion of cytokines (Table 1), activation of osteoclasts, decrease in osteoblasts and increase growth and multiplication of MM cells [20]. These processes cause angiogenesis, lytic bone lesions and drug resistance. The dysregulation of signaling pathways in diseased states is associated with the BM cells. These cells cause induction and activation of signaling pathways. The bone and stromal cells secrete soluble factors such as IGF-1, IL-6, IL-8, IL-17, TNF-α, SDF-1 and VEGF that promote diseased states (Figure 2) [21].

In solid tumors, the metastasis and growth of the tumor occur by neovascularization. However, in MM, despite being a nonsolid tumor, intensive neovascularization occurs, which results in the release of cytokines such as vascular endothelial growth factor (VEGF) and fibroblast growth factor-2 (FGF-2). MM cell growth is also promoted by IL-6, which is secreted by BM endothelial cells [21]. The plasma of MM patients contains elevated levels of VEGF in response to IL-6 activation by the plasma cells (Figure 1). The receptor of VEGF (VEGF-R), excessively expressed in MM cells, is considered to be involved in the autocrine signaling pathway [22]. VEGF is involved in the proliferation of MM cells by stimulating the MEK/ERK pathway and the migration of MM cells through PI3K pathway activation [23].

Moreover, it plays a role in the proliferation, through the mediation of the survival of MM cells by enhancing Mcl-1 and survivin expression. It increases bone resorption by osteoclast, causes angiogenesis (Figure 1), advances MGUS to the MM stage and raises microvessel marrow density (MVD). The bone damage occurs through the prevention of osteoblast formation by blocking the differentiation of mesenchymal stem cells, and it also occurs via osteoclast activation. These processes cause a reduction in bone formation and an increase in bone destruction. The growth and multiplication of MM cells in the BM are supported through the interaction of immunological, humoral, stromal and hormonal factors, as well as megakaryocytes and platelets [24]. Several novel drugs developed for MM patients based on the knowledge of the BM microenvironment target the BM microenvironment as well as the plasma cells [25].

Malignant cells affect the functionality of the BM microenvironment. Cell cycle and anti-apoptotic pathways, such as Janus kinase/signal transducer and activator of transcription 3 (JAK/STAT3), are activated due to the intercommunication of MM cells with BMSCs by adherence. This intercommunication of the cells causes upregulation of Bcl-xL and Mcl-1, which are anti-apoptotic proteins [26]. Malignant plasma cells also undergo metabolic changes, which affects several metabolic pathways and results in drug resistance in MM [27].

## 3. Role of Immune Checkpoints in MM

The immune system has a strong association with healthy bone maintenance. The inflammatory cytokines play a pivotal role in skeletal homeostasis, and their dysregulation can result in several adverse effects [21]. The pathogenesis of cancer consists of three stages: elimination, equilibrium and escape [28]. In the first stage, elimination, the immune system recognizes the cancer cells which are then eradicated through mechanisms such as immune cell cytolytic activity (i.e., NK cells, CD8+ T cells and γ δ T cells). It includes T cells, which are adaptive immune cells. [29,30]. In the second phase, equilibrium, a balance is established between cancer cell elimination and its proliferation through the regulation of control checkpoints. In the final stage, escape, if cancer persists, it overpowers the immune reactions and proceeds towards progression and metastasis. 

In the MM microenvironment, several immunosuppressive cytokines are produced, among which, IL-6 is of paramount importance. It is produced by MM cells, as well as BMSCs, and it inhibits the function of NK cells [31]. Moreover, NK cells, T cells and dendritic cells are also inhibited due to the production of TGF-β by stromal cells, MM cells and osteoblasts [32,33]. The genes which are involved in immunosuppression, such as IL-10, TGF-β and PD-L1, were found to be upregulated by APRIL (ligand of B-cell maturation antigen) [34]. Moreover, APRIL also induced the production of T cell regulators (Tregs) by MM cells and inhibited the T-cell proliferation through Tregs [32,35].

Due to the accuracy and long-term memory of immunotherapy, it is considered as a smart approach to attain durable remissions. Immune checkpoints play a vital role in maintaining homeostasis and are, hence, the immune system regulators. There are two classes of checkpoint molecules: stimulatory and inhibitory [36]. Stimulatory checkpoint molecules stimulate the immune system, such as CD27, CD28 and CD40, while inhibitory molecules inhibit it, such as CTLA4, KIRs and PD-1. Immune checkpoint inhibitory therapy is defined as a therapy that targets immune checkpoints. Immune checkpoints are the immune system’s crucial regulators, and they can activate or inhibit the action that cancer cells use to shield themselves from immune system attacks. Inhibitory checkpoints can be blocked by immune checkpoint therapy, which therefore restores the functions of the immune system. 

### 3.1. PD-1 Receptors

There is progressive immune dysfunction in myeloma, where BM milieu encourages immune evasion [37,38]. The PD-1/PD-L1 pathway plays a vital role in immune escape in MM, making this pathway of great interest [39]. Myeloma cells and many other cancer cells express PD-L1; however, healthy plasma cells do not express it. Moreover, it is also not expressed on plasma cells obtained from patients with MGUS [40,41,42,43]. IFN-γ and IL-6 are responsible for this increased PDL-1 expression in the MM microenvironment [44]. PD-L1 expression depends on the genetic subtype of the myeloma, which is expressed more in hyperdiploid cases as compared to nonhyperdiploid disease [45]. There is a reduced expression of PD-1 on circulating T cells obtained from patients who had the minimal disease after high-dose chemotherapy [42].

In contrast, there is a high expression of PD-1 on circulating t cells of patients with advanced MM [42]. In MM cells, there is an association of PD-L1 expression with drug resistance and increased proliferation [46]. PD-L1 is also considered to play a role in clonal resistance because of its increased expression on plasma cells obtained from patients with relapsed and refractory MM [47]. Furthermore, there is an increased PD-1 expression on NK cells derived from MM patients which inhibit the function of the effector cell [48]. This loss of function can be restored by blocking PD-1. Thus, the PD-1/PD-L1 pathway plays an essential role in immune escape in MM, and blocking this pathway can be a practical therapeutic approach [49]. However, the phase III clinical trials using anti-PD-1 mAb, pembrolizumab, have not been successful due to increased deaths in the intervention arm, and the trials were stopped by the Food and Drug Administration (FDA) (Table 2) [50].

In MM, besides the PD-1/PD-L1 pathway, some other processes, such as the induction of T-cell senescence [51] and CD226 [52], have also been involved in the reduction of tumor immunity. 

### 3.2. CTLA4 Receptors

A CTLA4 blocker, ipilimumab, is the first immune checkpoint therapy drug [53,54]. It is usually present at reduced concentrations on the surface of T cell effectors and Tregs. It functions in the activation of T cells during the initial stages [55]. CTLA4 and CD28 homodimerize and share the same ligands (CD80 and CD86) that are present on antigen-presenting cells (APCs). When CD28 binds with either CD80 or CD86 to deliver costimulation, the inhibitory CTLA-4 molecule shoots to the surface of the T cells. There, it binds with either CD80 or CD86 based on which has the higher affinity, consequently, counteracting the costimulatory CD28 activity through the binding of phosphatases pp2A and SHP2 [55,56]. CTLA4+ microvesicles, secreted by myeloid dendritic cells, have been shown to mediate immune suppression. Hence, mAbs can be used for CTLA-4 blockade to elevate the response of the immune system against cancer cells by inactivating Tregs and tumor-infiltrating lymphocytes (TILs) and by expanding the function of T helper (Th) cells [28]. 

CTLA-4 and PD-1 play a similar immunoregulatory role with slight, distinct differences concerning the regimes. CTLA-4 expression is limited to the subsets of T cells, but PD-1 expression is seen on activated B cells and natural killer cells (NK). Unlike CTLA-4, PD-1 recognizes two distinctly related ligands: PD-L1 and PD-L2. These ligands bind to the receptor in the setting of inflammation, and immune regulation is mediated by this interaction, which then protects against autoimmunity during the ongoing autoimmune response [45]. 

### 3.3. KIRs Receptors

Another type of receptors called KIRs belongs to a family of cellular receptors, which are mainly present on NK cells. They take part in both inhibiting and activating functions. It has been shown that the KIR ligand is disrupted to promote NK-cell alloreactivity. The most common inhibitory KIRs that are usually articulated on NK cells in myeloma are KIR2DL1, KIR2DL2 and KIR2DL3. A recent study showed that myeloma patients have increased the pervasiveness of KIR2DS4, which is a KIR that lacks function in several settings in comparison to healthy controls. Therefore, the KIR-ligand system might be one capable therapeutic aim for immune checkpoint inhibition in myeloma [45].

Although immune checkpoint inhibition therapy has promising results, there remain many unresolved challenges to overcome in the optimization of this approach in order to obtain maximum effectiveness. Therefore, further studies are required in this area of research to gain better outcomes.

## 4. Exosomes and microRNAs (miRNAs) in MM

Exosomes are a type of extracellular, membrane-bound vesicles (EVs). These vesicles are generated through endocytosis pathways where the cell membrane descends inside to create early endosomes, then late endosomes, and eventually develops into exosomes [57]. Approximately, all types of cells secrete exosomes and, in the microenvironment, they act as presenters of the signal. They contain microRNA (miRNA) and other biological information which can be transferred to the recipient cells [57]. In resistance of drug and tumor growth, exosome plays an essential role by delivering bioactive molecules such as cytokines, miRNAs, signaling molecules and growth factors [58]. Recipient cells take up the exosome through numerous ways: (1) phagocytosis, (2) target cell fusion with plasma membrane, (3) endocytosis mediated by receptor or lipiodraft or (4) micropinocytosis [57,58]. Exosomes can be considered as potential biomarkers for the diagnosis and classification of disease mainly due to the presence of miRNA [57,59]. The miRNA differs in number and types on different exosomes and different cells, and they are highly conserved in various species. The abnormal expression of these miRNA is associated with clinical diagnosis.

miRNAs are highly dysregulated in cancer [60]. miR-15a and miR-16 are tumor-suppressor miRNAs that are downregulated in refractory or relapsed MM (RRMM). These miRNAs are involved in the inhibition of growth and proliferation of MM cells both in vivo and in vitro. Also, the expression of these miRNAs downregulated the secretion of VEGF from MM cells and, therefore, reduced proangiogenic activity on endothelial cells [58,61,62]. Moreover, when normal cell lines/patients were compared with MM cell line/patients, the latter exhibited lower miR-133a, miR-124a, miR-125b, miR-16, miR-1 and miR-15 expression [63].

Various studies have revealed that abnormal regulation of miRNAs induces the stimulation of the PTEN-PI3K/AKT signaling pathway [64], associated with the development of bladder cancer, non-small cell lung cancer and breast cancer. As miRNAs play a significant role in many cancers, their modulation through therapeutics can serve as an essential approach in cancer study. miR-20a is known to work as an oncomir in several human cancers. It is upregulated in MM cells and also in the plasma from multiple myeloma patients [65]. The expression of miR-20a in the plasma of stage III patients was remarkably higher than in the patients of stages I and II, showing an essential role in the prediction of survival. The correlation between PTEN and miR-20a was also discovered, in which PTEN was seen to be targeted by miR-20a, and the overexpression of miR-20a suppressed the expression of PTEN.

Moreover, in MM cell lines, miR-20a serves as a PTEN negative regulator. In vivo, downregulation of miR-20a resulted in the MM cell growth inhibition [66]. Therefore, miR-20a is considered to play an essential role in MM through the modulation of PTEN-PI3K/AKT signaling [67]. 

In addition to the correlation of miR-20a and PTEN in the PTEN-PI3K/AKT signaling pathway, the relation of miR-21 and STAT3 has also been observed. The increased expression of miR-21 was seen in MM cells compared to the control cells, indicating its involvement in the pathogenesis of the disease. IL-6 has been seen to increase the oncogenic miR-21 levels through the STAT3 signaling pathway, which indicates that induction of miR-21 contributes to cancer progression through STAT3 [67,68].

## 5. Wnt/β Catenin Signaling Pathway and Its Association with MM 

Wnt signaling pathways maintain tissue homeostasis in mature organisms [69]. Dysregulation of Wnt signaling pathways lead to different human pathologies, particularly cancer [70]. Migration of cells during embryonic development, hematopoiesis and homeostasis of tissues are some of the functions that are regulated by the Wnt, which comprises two pathways: β-catenin-dependent pathway and the β-catenin-independent pathway. The β-catenin-dependent pathway functions in the regulation of the cell cycle, while the β-catenin-independent is involved in the regulation of intracellular calcium and planar cell polarity [71].

Interruptions in the Wnt pathway, mainly in the β-catenin-dependent pathway, are linked to cancer and a series of other developmental diseases. In the absence of the Wnt ligand, β-catenin is phosphorylated and marked for proteasomal degradation by the destruction complex, which consists of GSK3β, AXIN, APC and CK1α. On the other hand, when the Wnt ligand binds to its receptor, the docking site for AXIN is formed by the phosphorylated co-receptors, LRP5/6. Subsequent removal of AXIN results in disruption of the destruction complex, which allows translocation of nonphosphorylated β-catenin to the nucleus (Figure 3) [72]. In the nucleus, β-catenin binds to T-cell factor/lymphoid enhancer factor 1 (TCF/LEF1) [70]. The establishment of this transcription factor complex causes the transcription of several genes. In addition to the cytoplasmic pool, β-catenin binds to E-cadherin (ECAD) on the cytoplasmic domain. ECAD is a protein that functions in intracellular adhesion. The gene CDH1 codes for ECAD, and the loss of this gene is related to the increased invasion of tumors and contributes to metastases. 

It has been shown that β-catenin/transforming growth factor (TGF)/LEF1 functions in transcriptionally activating CCDN1 and MYC, which are upregulated in MM (Table 1) [73,74]. The β-catenin/TGF/LEF complex downregulates p16INK4a and miR-15a/16 expressions, which further cause cyclin D1 upregulation and enhance angiogenesis in MM [61,75,76]. 

### Inhibitors of Wnt Signaling 

Proteins that inhibit the Wnt pathway are divided into two classes: the secreted Frizzled-related protein (sFRP) class and the Dickkopf (Dkk) class [77]. The sFRP class consists of the sFRP family, Cerberus and Wnt inhibitory factor 1 (WIF-1); all of these directly bind to Wnt ligands. Whereas, the Dkk class comprises of the Dkk protein family, which inhibits the Wnt pathway by binding to LRP5/6. Besides this, it has been shown that miR-34a, MiR-203, miR-21 and miR-200a are inhibitors of the Wnt pathway. In MM, miR-34a and MiR-203 are downregulated, while miR-21 and miR-200a are upregulated [73].

The antagonists of Wnt participate indirectly in tumor growth by reducing the differentiation of osteoblasts. The osteoblast precursors support the growth of tumors by secreting higher levels of growth factors and cytokines such as VEGF, HGF, BAFF, IL-6 and IL-10, which suggests that inhibition of osteoblast differentiation is involved in tumorigenesis [78,79,80]. The impairment of osteoblast differentiation also forms osteolytic lesions due to a Wnt antagonist (Figure 2) [72].

## 6. Role of PTEN-PI3K/AKT Signaling in MM

The PTEN-PI3K/AKT pathway is known as a significant molecular pathway in the progression and development of malignant cells [81,82]. This pathway is highly activated in the cancer cells; thus, it is involved in drug resistance, growth, survival, invasion and migration of cancer cells (Table 1) [66,83,84]. 

The PI3K/AKT pathway starts with the activation of PI3K via three different pathways, two of which are stimulated by the activation of one of the tyrosine kinases receptors through an extracellular growth factor, which causes the dimerization and phosphorylation of the receptor monomers. Different proteins tend to bind to the receptor’s phosphorylated domain, which functions as a binding and activation site for the PI3K. Besides, PI3K can get activated by binding directly to one of the tyrosine kinase receptors. The third pathway begins with a Ras protein, in which PI3K gets activated by binding to an active Ras protein. The active PI3K moves inside the cell membrane and gets attached to phosphatidylinositol (4,5)-bisphosphate (PIP2), a common element of the cell membrane that is held by two fatty acids in the lipid layer of the membrane. PI3K phosphorylates PIP2 to PIP3, which then activates the kinase AKT. AKT plays a pivotal role in the downregulation of apoptosis, as it activates protein formation or translation and inhibits BAX proteins from binding to the mitochondrial membrane to prevent apoptosis [85].

PTEN (cancer suppressor gene) is a vital signal protein in the PI3K/AKT pathway that functions in dephosphorylating lipid phosphatidylinositol-3,4,5-triphosphate (PIP3). PIP3 is a product of PI3K that accumulates when PI3K is overactivated [86]. 

The targeting of the PI3K/AKT pathway in MM is an important therapeutic choice, because this pathway plays a vital role in the pathogenesis and drug resistance of MM. It has been shown that the growth and progression of MM cells are significantly reduced in vitro and in vivo when AKT is inhibited by TAS-117, which is a potent and selective allosteric inhibitor of AKT [87]. TAS-117 results in a stress response in the endoplasmic reticulum (ER), as well as autophagy and apoptosis. Moreover, it was found that TAS-117 inhibits the secretion of IL-6 from the BMSCs by regulating the activity of NF-κB [83]. Hence, the signaling pathway PTEN-PI3K/AKT is considered as an essential pathway for drug discovery, because it plays a vital role in cell proliferation, cell growth, apoptosis and metabolism. 

**Table 1 ijms-21-05002-t001:** The role of dysregulated signaling pathways in multiple myeloma (MM) pathogenesis.

Signaling Pathways	MM Pathogenesis	Factor of Solubility	Role in MM Pathogenesis	Affected Gene	References
Wnt/β-catenin	Apoptosis Proliferation Migration Differentiation Self-Renewal Osteolytic bone disease	IL-6	The Wnt/β-catenin promotes signaling and cell growth in MM.	cyclin D1 c-myc axin-2	[72,79,88,89,90,91]
PI3K/Akt/mTOR	Survival Apoptosis Proliferation Resistance Migration Angiogenesis	IL-6	The PI3K/Akt/mTOR pathway is involved in growth, survival and drug resistance in MM plasma cells. This signaling pathway is also involved in managing establishment and activity of osteoblasts and osteoclasts.	Gene 6 (Gas6)	[92,93,94,95,96,97]
NF- κB	Survival Apoptosis Proliferation Angiogenesis Metastasis Osteolytic bone disease	IL-6 IL-8 APRIL	NF- κB is an important transcription factor that regulates cell survival in various cells. When it is activated, it protects some hematopoietic neoplastic cells from apoptosis.	Gene 6 (Gas6)	[94,98,99,100,101,102,103]
JAK/STAT	Survival Apoptosis Proliferation Resistance Migration Differentiation Transformation	IGF-1 IL-6	The JAK/STAT pathway plays a pivotal role in the organization of the immune system, particularly cytokine receptors, and they can change the polarization of T helper cells.	SHP-1 SHP-2 SOCS-1	[104,105,106,107,108,109]
Ras/Raf/MEK/Erk	Survival Proliferation Migration Angiogenesis	VEGF IL-10	The RAS/MEK/ERK pathway is considered to be activated in about half of MM cases and is, therefore, a primary therapeutic target in MM.	HSV-2 gene ICP10PK	[110,111,112,113,114]
RANK/RANKL/OPG	Osteolytic bone disease	HGF	RANK/RANKL/OPG play an essential role in bone metabolism, affecting osteoclast formation and activity.	-	[115,116]

## 7. Role of JAK/STAT Signaling and Suppressors of Cytokine Signaling (SOCS) in MM

Janus-associated kinase-signal transducer and activator of transcription (JAK-STAT) pathway is a principal signaling cascade that controls the expression of genes by transferring extracellular signals to the nucleus [117]. The JAK family consists of four tyrosine kinases that are: Janus kinase 1–3 (JAK1, JAK2 and JAK3) and tyrosine kinase 2 (TYK2). These kinases bind to cytokine receptors in the juxta-membrane region. The kinase activity of JAK is initiated when the ligand is attached to its receptor in the process of receptor dimerization [118]. The cytoplasmic region of the receptor is phosphorylated through subsequently activated JAKs. The activated complex of the JAK-cytokine receptor phosphorylates and recruits specific transcription factors, STAT proteins. Seven STAT family proteins found in human cells are STAT1-6, in which STAT5 includes STAT5a and STAT5b [117]. The phosphorylated STAT proteins form dimers and move to the nucleus where they function in gene transcription by interacting with several regulatory elements [119].

In the myeloma cells and the myeloma cell lines obtained from patients, STAT3 has been seen to be active, yet it remains inactive in the plasma cells obtained from healthy individuals [120,121]. IL-6 is a stimulator of STAT3 in myeloma. When IL-6 binds to its receptor (IL-6R), it causes the activation of tyrosine kinases that belong to the JAK family of proteins [122]. STAT3 plays a significant role in the survival, proliferation and chemoresistance of MM cells [122]. Due to the function of IL-6 as a proliferative factor in MM cells, its inhibition was considered as a therapeutic strategy for MM [10]. 

### SOCS

The regulation of several biological functions (such as healing of wounds and immunity) is carried out by cytokines, which are also crucial for commencing the JAK/STAT pathway protein. The JAK/STAT pathway, facilitated by cytokine signaling, plays a vital role in processes that are involved in cancer initiation and growth, such as multiplication, maturation, differentiation and apoptosis of various cell types [123,124]. In mammals, the SOCS family contains eight members that are: cytokine-induced SH2 containing protein (CIS) and SOCS1-7. SOCS 1 and 3 are predominantly effective inhibitors of the JAK/STAT pathway (Figure 3) [125]. Cytokine signals are inhibited by SOCS protein in four ways, which are: (1) Block the recruitment of STATs to the cytokine receptor by masking the STAT’s receptor binding sites. (2) Target proteins for degradation by proteasome through ubiquitination. (3) Bind to JAKs and inhibit their kinase. (4) Target the degradation of JAKs through the proteasome. Based on observation, SOCS1 expression is decreased in numerous cancers, such as prostate cancer, MM, lymphoma, hepatocellular carcinoma, acute myeloid leukemia, pancreatic cancer and laryngeal carcinoma [126,127]. 

The persistent activation of STAT3 has been found in many tumor cells, such as breast cancer, head and neck cancer [128], colorectal cancer, hepatocellular carcinoma [129], renal cell carcinoma, prostate cancer, ovarian cancer [130] and leukemia [131]. The expression level of SOCS3 has been shown to be reduced in cancerous lesions that are infected with HCV compared to noncancerous lesions. Reduced levels of SOCS3 may cause the hyperactivation of STAT3, which induces multiple tumor-promoting genes and, hence, contributes to malignancies and carcinogenesis.

Suppressing cytokine signaling by using SOCS can be a useful therapy in cancer treatment [125]. One such way for treatment is the suppression of the tumor that promotes STATs by overexpressing the SOCS protein, which will inhibit the growth of the tumor. Another way is through anticancer immunity enhancement by SOCS silencing in the dendritic cells or tumor cell lysates.

It has been shown that in vivo and in vitro SOCS1 overexpression inhibits the growth of human melanoma cells. Furthermore, SOCS1 is associated with Cdh1, which initiates the degradation by proteasomes [132]. In non-small lung cancer cells, growth is inhibited by the overexpression of SOCS3. Also, the radio-sensitivity of treated non-small lung cancers cells is enhanced by the overexpression of SOCS3. Hence, SOCS 1 and 3 or SOCS-mimetic overexpression can become useful therapy in cancer treatment. Based on the JAK/SOCS complex structural analysis, SOCS development is highly desirable [125]. 

## 8. Role of Nuclear Factor-kappaB (NF-κB) in MM 

Nuclear factor kappa-light-chain enhancer of activated B cells (NF-κB) is detected in the cytosol in most of the cell types. It inhibits apoptosis and promotes inflammation, metastasis and the proliferation of cells [133]. It is nonfunctional when it is associated with inhibitors of the IκB family [134,135]. NF-κB is activated when activators such as TNF-α activate IκB kinase (IKK), which phosphorylates and degrades IκB through the 26S proteasome. NF-κB then moves to the nucleus from the cytoplasm, where it starts the transcription of the target genes [122,136]. 

NF-κB is a complex of proteins that play a role in the immunity, transcription of DNA, inflammation and cell survival [137]. NF-κB works in many cells, and it coordinates adaptive and innate immune responses. However, dysregulation of NF-κB signaling has been reported in many human malignancies, including cancers [138]. The NF-κB family has five monomeric subunits which are structurally somewhat similar: RelA (also known as p65), RelB, c-Rel, p50 (produced as a p105 precursor protein and encoded by NF-κB1) and p52 (produced as p100 precursor protein and encoded by NF-κB2) [2]. The mature subunits associate with each other to create 15 possible hetero or homodimeric transcription factors, out of which, the most common dimers are RelB:p52 and RelA:p50. The N-termini of NF-κB proteins contains a conserved Rel homology region (RHR), which possesses a nuclear localization sequence as well as domains for DNA-binding and dimerization. NF-κB dimers recognize the κB motif, a large DNA sequence. In resting cells, the inhibitor proteins keep the NF-κB factors in their inactive form in the cytoplasm [2]. The two NF-κB pathways, canonical and noncanonical, are triggered by extracellular stimuli to move NF-κB dimers into the nucleus where they initiate the expression of hundreds of stress response and immune response genes, as well as pro-survival and immune-differentiating factors [2].

The canonical and noncanonical pathways differ in biological functions and signaling components [139]. The canonical NF-κB pathway’s activation mainly relies on the degradation of IκBs, specifically IKBα, which is mediated by the IκB kinase through phosphorylation [140,141]. The IKK phosphorylates IκB-specific and terminal serine residues. The inhibitory subunits IkBα, IkBβ and IkBε undergo proteasomal breakdown after phosphorylation [142]. Therefore, the NF-κB dimer gets transferred to the nucleus. The noncanonical NF-κB pathway is primarily involved in B cell maturation, osteoclast differentiation, lymphoid organogenesis and immune system functioning. In the inactivated form of the NF-κB complex, there are a series of proteins which are bound to each other and function in the proteasomal degradation of NF-κB-inducing kinase (NIK).

### Interleukins and Growth Factors

BMSCs activate the canonical NF-κB pathway upon adhering to MM cells, which induces IL-6 expression [143]. Moreover, the IL-6 production is also caused by IL-1β through the induction of the canonical NF-κB pathway in BMSCs. IL-6 induces the production of VEGF and also exerts pro-survival and pro-proliferative gene expressions in MM cells. NF-κB-target genes also encode some of the VEGF isoforms. Furthermore, the NF-κB dependent antiapoptotic gene expression is caused by IGF-1, which is secreted by BMSCs [144,145]. The canonical NF-κB pathway is activated by TNF (a pro-inflammatory cytokine) in both myeloma cells as well as in BMSCs, and it is also secreted by the canonical pathway [2]. NF-κB promotes the downstream signaling of the receptors that is initiated by the growth factors, e.g., epidermal growth factor receptor (EGFR). Due to the activation of EGFR, P13K is released in the plasma membrane, which causes the production of PIP3. [146]. PIP3, with the help of phosphoinositide-dependent kinase-1 (PDK1) and the mammalian target of rapamycin complex 2 (mTORC2), activates protein kinase B (AKT/PKB). The mutations that affect mTOR and EFGR are particularly crucial in activating AKT, which in turn activates NF-κB through IKK. This activation of NF-κB results in the transcription of prosurvival genes and, thus, inhibits cancer cell death.

There are many treatment strategies for MM patients, such as histone deacetylase inhibitors or proteasome inhibitors and immunomodulatory drugs [147]. Besides these, there are some novel therapies, which include chimeric antigen receptor T cells (CAR-T cells) and small molecule inhibitors. Table 2 discusses the mechanism of action of some drugs alone, or in combination with other drugs, on MM patients.

The NF-κB pathway, besides being essential for normal lymphoid cells, is also required for the survival and proliferation of B-cell acquired cancers, including MM [147]. Thus, the NF-κB signaling pathway is the primary or secondary target of many compounds which are used to treat MM [147].

## 9. Role of FOXM1 in Drug Resistance of MM

Forkhead box M1 (FOXM1) regulates numerous biological processes which are all associated with the progression of tumors and their reaction to targeted therapies [148]. These processes involve the repair of DNA damage, cell cycle progression, stem cell self-regeneration and senescence [149]. FOXM1 is involved in tumor cell proliferation and is overexpressed in numerous types of cancers [150]. In vitro, the upregulation of FOXM1 has been shown to increase drug resistance in myeloma [148]. In MM cells, this transcription factor supports the progression of the cell cycle and also interacts with the cyclin D-CDK4/6-Rb-E2F pathway to promote drug resistance [148]. 

Regarding the various biological roles ascribed to the FOXM1 in MM, it resembles some well-recognized transcription factors of myeloma, such as myelocytomatosis oncoprotein (MYC) and interferon regulatory factor 4 (IRF4) [151]. FOXM1 is also involved in the progression of other B-lineage neoplasms, which include chronic lymphocytic leukemia [152], acute lymphoblastic leukemia [153], follicular lymphoma [154] and diffuse large cell lymphoma [155]. According to a new pan-cancer meta-examination of nearly 18,000 gene expression signatures, FOXM1 is recognized as a significant predictor of detrimental outcomes in 39 hematologic and solid malignancies, including MM. [156].

In newly diagnosed myeloma patients, the FOXM1 gene is associated with high risk [157], which, upon tumor relapse, experiences further increases in most cases [156]. FOXM1’s interaction with NIMA-related kinase 2 (NEK2) and the CDK4/6-Rb-E2F axis, in myeloma cells, is considered to be useful from a therapeutic aspect, because CDK inhibition is thought to be effective in myeloma treatment [156,157,158,159]. Furthermore, in cancer, NEK2 is targeted by FOXM1, and it has been shown to cause drug resistance in MM and some other malignancies [160,161]. Small compounds that exhibit the properties to inhibit kinase activity can be used to target it [162] or can indirectly cause deterioration of a target by a process that involves an interruption in the binding of NEK2 to the NDC80/HEC1 component of the kinetochore complex [148].

Drug resistance is the primary concern in cancer treatment [163]. In drug-sensitive cells of breast cancer, FOXM1 is downregulated by chemotherapy, but in the resistant cells, FOXM1 maintains its levels [164]. Among these chemotherapy agents, cisplatin, which is a highly effective drug, forms adducts of platinum on genomic DNA, causing the damage of DNA and, eventually, cell death [163]. It has been shown that, in the cell line MCF-7, treatment with cisplatin activated the repair of DNA in the resistant MCF-7-CISR as compared to MCF-7 cells. The active FOXM1 expression in cisplatin-sensitive MCF-7 cells also contributes resistance, but FOXM1 silencing can make the cells sensitize to the drug [164]. FOXM1 confers resistance in cancer cells by increasing DNA damage repair; thus, the drugs targeting FOXM1 can prove promising results in inducing cell death in resistant tumor cells [163]. 

RNA interference can be used as a strategy to downregulate FOXM1, which eventually inhibits the invasion and proliferation of cancerous cells. The transcription factor FOXM1 is considered a direct target of thiostrepton, which is a proteasome inhibitor, and blocks the binding of FOXM1 to its target genomic sequence, thus abort its transcriptional activity [165]. The overexpression of FOXM1 is linked with the expression of VEGF and MMP-9. VEGF and MMP-9 are the proteins that are involved in the angiogenesis of cancer and in the degradation of the ECM. In a study conducted by Siraj A K et al. [166], it was shown that treatment of breast cancer cells with thiostrepton noticeably downregulated FOXM1, MMP-2, MMP-9 and VEGF expression. 

**Table 2 ijms-21-05002-t002:** Mechanism of action and/or toxicities of different drugs in MM patients.

Drug/Antibody	Model/System	Mechanism of Action/Toxicity	Reference
Pembrolizumub(Phase III clinical trial)	301 newly diagnosed MM patients (NDMM)	The adverse events in patients included cardiac arrest, cardiac failure, myocarditis, pneumonia, intestinal ischemia, pulmonary embolism, cardiorespiratory arrest, sepsis and large intestinal perforation	[167]
Pembrolizumub (phase III clinical trial)	249 patients having (RRMM)	Adverse reaction of the drug included pericardial hemorrhage, neutropenic sepsis, Steven-Johnson syndrome, myocarditis, respiratory tract infection, sepsis, cardiac failure and myocardial infarction.	[168]
Ixazomib (phase III clinical trial)	722 patients having RRMM	Ixazomib is a proteasome inhibitor that binds to the 20S proteasome at β5 subunit and inhibits its activity. It reduces the release of cytokines by inhibiting the NF-κB pathway in vitro in multiple myeloma stromal cells.	[169]
Thalidomide (phase II clinical trial)	34 patients, previously treated with three or fewer therapies	Thalidomide inhibits the production of TNF-α by monocytes. It also inhibits the survival of MM cells by affecting the BM microenvironment.	[170,171]
Isatuximab in combination with Pomalidomide/dexamethasone	307 patients who had RRMM	Isatuximab is an IgG1 κ mAb. It binds to CD38 on a unique site and kills tumor cells through different mechanisms such as phagocytosis and cellular cytotoxicity. Cellular cytotoxicity, which is mediated by NK cells, is the highly effective mechanism caused by isatuximab.	[172,173]
AMG 424	Mice and cynomolgus monkeys	This antibody causes multiplication of T cells and is targeted at CD38, which is a cell surface marker of MM. It results in the complete killing of myeloma cells that express high and low levels of CD38.	[174]
Daratumumab	53 patients with RRMM	Daratumumab is a human monoclonal IgG kappa antibody which targets CD38. The adverse events associated with daratumumab include some hematological toxicities.	[175]
Lenalidomide (LEN) in combination with dexamethasone (DEX)	98 patients were treated with one cycle of LEN/DEX and 48 patients with 6 cycles	Lenalidomide is an immunomodulatory drug, and it possesses apoptotic and antiangiogenic properties.	[176,177]
Elotuzumab/immunomodulatory drug and dexamethasone	33 patients with RRMM (safety and efficacy of elotuzumab)	Elotuzumab causes activation of NK cells by SLAMF7 ligation to exert its anti-myeloma function. Moreover, it also results in antibody-dependent cellular cytotoxicity. The frequent adverse event recorded was lymphopenia.	[178]
Carfilzomib (pooled analysis of phase I and II studies)	121 newly diagnosed MM patients were analyzed, those who were transplant-ineligible	Carfilzomib is a proteasome inhibitor (PI). PIs mainly target the 20S proteasome, which is involved in the proliferation of cancerous cells.	[179,180]
MOR202	38 RRMM patients (safety and efficacy of MOR202)	MOR202 is a HuCAL-derived, anti-CD38 monoclonal antibody which shows effective cellular cytotoxicity, cell-mediated phagocytosis and significant activity in preclinical multiple myeloma models. MOR202 was found to be safe and well-tolerated.	[181]
Bortezomib, dexamethasone and lenalidomide (phase II clinical trial)	64 patients with RRMM	Preclinical studies in MM cells have demonstrated that the activity of dexamethasone is stimulated by bortezomib, and the introduction of lenalidomide makes the cells sensitive to bortezomib, suggesting that the combination of these proteasome inhibitors and immunomodulatory drugs can enhance patients’ survival rates.	[182]

## 10. Conclusion and Future Perspective 

MM remains an incurable malignancy owing to drug resistance, which requires further investigation to find a cure. The changes in the expression of cytokines and adhesion molecules are major reasons for dysregulated signaling pathways in tumor cells. Therefore, it seems likely that the inhibition of cytokine-dependent MM signaling pathways can help in overcoming drug resistance. Moreover, the thorough understanding of the association between inflammation and MM can help in developing effective therapeutic interventions. 

Over the last decade, several new therapies for the treatment of myeloma have been considered and investigated. The new therapies not only include novel drugs but antibodies and vaccine-based therapies and new immune and cellular approaches as well. Theoretically, the combination of drugs with cellular and immune therapies may have better outcomes for MM patients, like a disease-free, long-term survival.

## Figures and Tables

**Figure 1 ijms-21-05002-f001:**
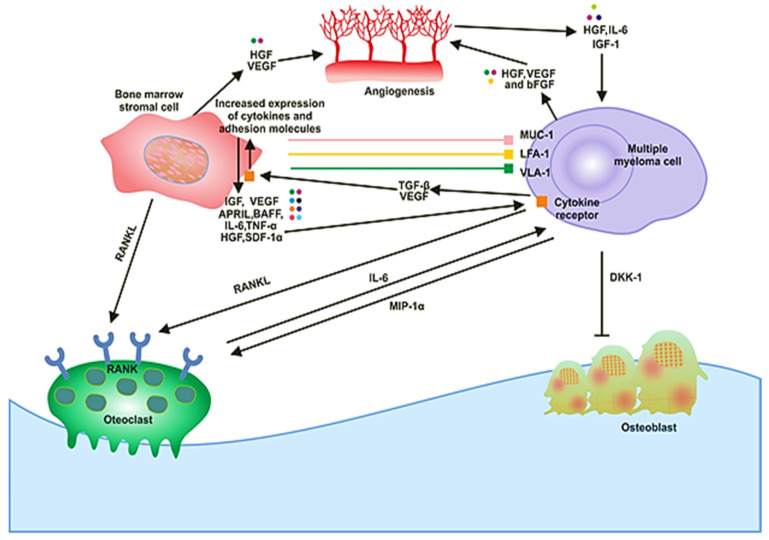
The signaling and cellular interactions between multiple myeloma (MM) cells and the bone marrow microenvironment.

**Figure 2 ijms-21-05002-f002:**
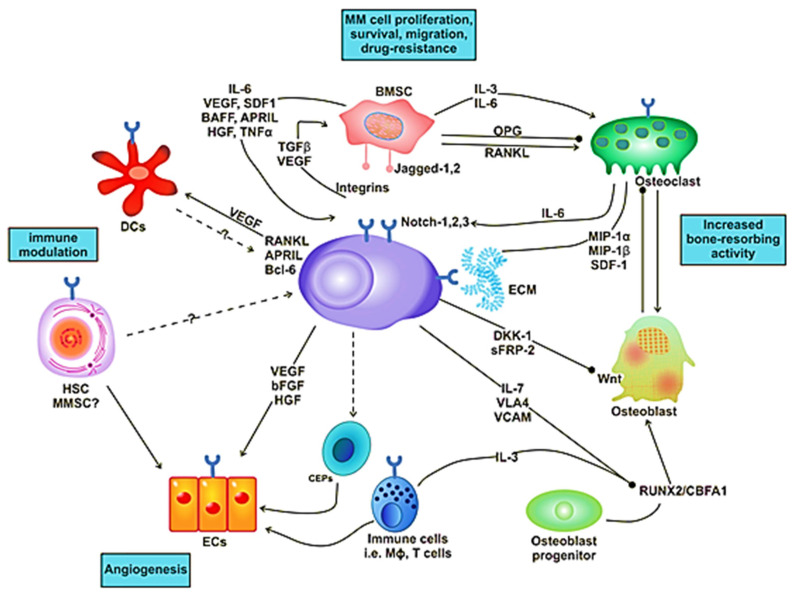
Interaction of MM cell with different compartments of the bone marrow (BM) microenvironment.

**Figure 3 ijms-21-05002-f003:**
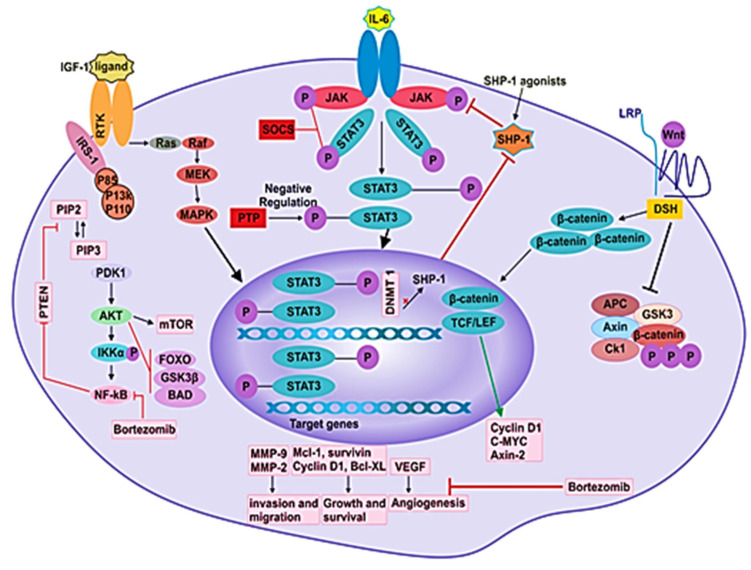
Different signaling pathways in MM pathogenesis. Signaling pathways such as JAK/STAT, PI3K/Akt, NF-kB, Ras/Raf/MEK/Erk and Wnt/β-catenin participate in the pathogenesis of disease by mediating the angiogenesis, proliferation, survival, differentiation, invasion and migration of MM cells.

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
