# Peer review of "Cytokine-Mediated Dysregulation of Signaling Pathways in the Pathogenesis of Multiple Myeloma"

_ijms, 2020, doi:10.3390/ijms21145002_

Round 1

Reviewer 1 Report

The authors described roughly the main mechanisms underlying the pathogenesis of multiple myeloma. there are a number of these reviews in literature. It should be more interesting to expande the review with new aspects of the topic eg. the role of exosomes and miRNA. Moreover, other cells are clearly involved in the pathogenesis of the disease such as osteocytes and their relationship with MM cells during bone loss. Please include this section.  The authors cited the involvment of inflammation in the pathogenesis of MM without considering the main cytokines involved (interleukines)

Author Response

The authors described roughly the main mechanisms underlying the pathogenesis of multiple myeloma. there are a number of these reviews in literature. It should be more interesting to expand the review with new aspects of the topic eg. the role of exosomes and miRNA. Moreover, other cells are clearly involved in the pathogenesis of the disease such as osteocytes and their relationship with MM cells during bone loss. Please include this section. 

Response: We appreciate the reviewer’s feedback and suggestions. In our review, we are mainly focusing on the role of different cytokines in the dysregulation of cell signalling pathways. We had earlier discussed the role of microRNA, but it was a short section. Now we have included a separate section about the role of exosomes and miRNA as per reviewer’s suggestion. Moreover, in the manuscript, we have also discussed that osteoclasts and osteoblasts are affected in the different signalling pathways.

The authors cited the involvement of inflammation in the pathogenesis of MM without considering the main cytokines involved (interleukins)

Response:  We have discussed almost every role of cytokines about specific signalling pathways. We believe that this suits the theme of our manuscript.

Reviewer 2 Report

The manuscript by Sabah Akhtar reviews all inflammatory pathways in the context of multiple myeloma. The manuscript has a lot of information but needs to be rewritten and re-structured to improve it.

Manor changes:

Along the manuscript there is quite a lot of repeated or redundant information.  All the manuscript should be revised and re-structured to eliminate all redundant information.

PD1-PDL1 section: It should be mentioned that clinical trials in MM patients with anti-PD1 have not been successful due to their high toxicity. Mention these clinical trials in Table 2.

Minor changes:

All figures need to have higher resolution and increase the font size of all the names of genes in the figures.

Include the borders of the tables to make them easier to read.

Along the manuscript there are many gene names which appear abbreviated and afterwards they appear with the complete word. The first time that the gene name appears should be written with the full name and in brackets the abbreviated form, and afterwards all the times that appears in the abbreviated form.

Line 260: “Deterioration”: For the proteasome is better degradation than deterioration

Line 270-272: it is not clear the meaning.

Line 345: mutation of EFGR: not clear the meaning in the context of the sentence.

Line 363: “In respect”: what does it mean? it might not be better regarding?

Line 442: “an expression of PD1”: change for high expression of PD1.

Line 455: CTLA4 and CD28 (CD80 and CD86): it is not clear the meaning.

Line 456: APC (antigen presenting cells). It should be antigen presenting cells (APC).

Lines 466: appear additional information again with PD1. This information should be included earlier in the PD1 section.

Author Response

The manuscript by Sabah Akhtar reviews all inflammatory pathways in the context of multiple myeloma. The manuscript has a lot of information but needs to be rewritten and re-structured to improve it.

Response: We appreciate the reviewer’s valuable feedback and a very constructive suggestion. We have restructured and rewritten a few sections in the revised our manuscript.

PD1-PDL1 section: It should be mentioned that clinical trials in MM patients with anti-PD1 have not been successful due to their high toxicity. Mention these clinical trials in Table 2.

Response: The reviewer has raised a very valid point, and we appreciate it.  As per reviewer suggestion we have mentioned in the revised manuscript that the clinical trials in MM patients with anti-PD1 have not been successful and also included them in Table 2.

Minor changes:

All figures need to have higher resolution and increase the font size of all the names of genes in the figures.

Response:  This point is well taken, and We have increased the resolution of all figures in the revised manuscript.

Include the borders of the tables to make them easier to read.

Response: We have included the borders of the tables.

Along the manuscript there are many gene names which appear abbreviated and afterwards they appear with the complete word. The first time that the gene name appears should be written with the full name and in brackets the abbreviated form, and afterwards all the times that appears in the abbreviated form.

Response: We appreciate the above comments. We have modified all the gene names by writing the full form first, which is followed by the abbreviated form and using the abbreviated form afterwards in the revised manuscript.

Line 260: “Deterioration”: For the proteasome is better degradation than deterioration

Response: Deterioration has been changed to degradation.

Line 270-272: it is not clear the meaning.

Response: The sentence has been deleted.

Line 345: mutation of EFGR: not clear the meaning in the context of the sentence.

Response: The sentence has been modified.

Line 363: “In respect”: what does it mean? it might not be better regarding?

Response: It has been changed to ‘regarding’.

Line 442: “an expression of PD1”: change for high expression of PD1.

Response: The sentence has been modified.

Line 455: CTLA4 and CD28 (CD80 and CD86): it is not clear the meaning.

Response: The sentence has been modified.

Line 456: APC (antigen presenting cells). It should be antigen presenting cells (APC).

Response: The sentence has been modified.

Lines 466: appear additional information again with PD1. This information should be included earlier in the PD1 section. 

Response: We appreciate that the reviewer has highlighted this. There is a single sentence that compares PD1 receptors and CTLA4 receptors, and we believe that this comparison should come after the CTLA4 section. The additional information about PD1 has been moved to the PD1 section.

Round 2

Reviewer 1 Report

thank you for your reply. I appreciate the new sections

Author Response

Reviewer comments

thank you for your reply. I appreciate the new sections

Authors Response: We are very thankful to reviewer for the positive feedback

Reviewer 2 Report

The manuscript has improved substantially. A few minor changes related to style need to be done:

Minor changes:

Line 125: “In the first stage, elimination, the innate immune system recognizes the cancer cells which are then eradicated through mechanisms such as immune cell cytolytic activity (i.e., NK cells, CD8+ T cells, and γ δ T cells)”. It includes T cells which are adaptive immune cells. Therefore, delete “innate” from innate immune system.

Line 158-159: “There is a reduced expression of PD-1 on circulating T cells obtained from patients who had the minimal disease after high-dose chemotherapy”. Please, cite reference.

Line 168-169: “pembrolizumab, has not been successful due to increased deaths in the intervention arm due to which the trials were stopped by food and drug administration (FDA)”. It is not correct the expression due to which the trials were stopped. May be: …and the trials were stopped by FDA?

Line 235: “Moreover, in the MM cell line”: replace by “Moreover in MM cell lines”.

Line 241: “The IL-6”. Replace by IL6.

Table 2: “mice and cynomolgus monkeys”: Write Mice instead of mice (capitals).

Author Response

Reviewer 2 Comments

The manuscript has improved substantially. A few minor changes related to style need to be done:

Authors Response: We are thankful to reviewer and editorial board for a positive feedback. The quality of article has been improved after adding the suggestion raised by esteemed reviewer.

Minor changes:

Line 125: “In the first stage, elimination, the innate immune system recognizes the cancer cells which are then eradicated through mechanisms such as immune cell cytolytic activity (i.e., NK cells, CD8+ T cells, and γ δ T cells)”. It includes T cells which are adaptive immune cells. Therefore, delete “innate” from innate immune system.

 Authors Response: We have deleted innate.

Line 158-159: “There is a reduced expression of PD-1 on circulating T cells obtained from patients who had the minimal disease after high-dose chemotherapy”. Please, cite reference.

 Authors Response: We have cited the reference in the revised manuscript.

Line 168-169: “pembrolizumab, has not been successful due to increased deaths in the intervention arm due to which the trials were stopped by food and drug administration (FDA)”. It is not correct the expression due to which the trials were stopped. May be: …and the trials were stopped by FDA?

 Authors Response: We have edited as suggested by the reviewer.

Line 235: “Moreover, in the MM cell line”: replace by “Moreover in MM cell lines”.

 Authors Response: We have modified it.

Line 241: “The IL-6”. Replace by IL6.

 Authors Response: It has been edited in the revised manuscript

Table 2: “mice and cynomolgus monkeys”: Write Mice instead of mice (capitals).

Authors Response: We have edited it in the revised manuscript.